# Incremental Few-Shot Learning with Attention Attractor Networks

**Mengye Ren**[1,2,3]**, Renjie Liao**[1,2,3]**, Ethan Fetaya**[1,2]**, Richard S. Zemel**[1,2]
[1]University of Toronto, [2]Vector Institute, [3]Uber ATG
{mren, rjliao, ethanf, zemel}@cs.toronto.edu

## Abstract

Machine learning classifiers are often trained to recognize a set of pre-defined classes. However, in many applications, it is often desirable to have the flexibility of learning additional concepts, with limited data and without re-training on the full training set. This paper addresses this problem, *incremental few-shot learning*, where a regular classification network has already been trained to recognize a set of base classes, and several extra novel classes are being considered, each with only a few labeled examples. After learning the novel classes, the model is then evaluated on the overall classification performance on both base and novel classes. To this end, we propose a meta-learning model, the Attention Attractor Network, which regularizes the learning of novel classes. In each episode, we train a set of new weights to recognize novel classes until they converge, and we show that the technique of recurrent back-propagation can back-propagate through the optimization process and facilitate the learning of these parameters. We demonstrate that the learned attractor network can help recognize novel classes while remembering old classes without the need to review the original training set, outperforming various baselines.

## 1 Introduction

The availability of large scale datasets with detailed annotation, such as ImageNet [30], played a significant role in the recent success of deep learning. The need for such a large dataset is however a limitation, since its collection requires intensive human labor. This is also strikingly different from human learning, where new concepts can be learned from very few examples. One line of work that attempts to bridge this gap is few-shot learning [16, 36, 33], where a model learns to output a classifier given only a few labeled examples of the unseen classes. While this is a promising line of work, its practical usability is a concern, because few-shot models only focus on learning novel classes, ignoring the fact that many common classes are readily available in large datasets.

An approach that aims to enjoy the best of both worlds, the ability to learn from large datasets for common classes with the flexibility of few-shot learning for others, is *incremental few-shot learning* [9]. This combines incremental learning where we want to add new classes without catastrophic forgetting [20], with few-shot learning when the new classes, unlike the base classes, only have a small amount of examples. One use case to illustrate the problem is a visual aid system. Most objects of interest are common to all users, e.g., cars, pedestrian signals; however, users would also like to augment the system with additional personalized items or important landmarks in their area. Such a system needs to be able to learn new classes from few examples, without harming the performance on the original classes and typically without access to the dataset used to train the original classes.

In this work we present a novel method for incremental few-shot learning where during meta-learning we optimize a regularizer that reduces catastrophic forgetting from the incremental few-shot learning. Our proposed regularizer is inspired by attractor networks [42] and can be thought of as a memory of the base classes, adapted to the new classes. We also show how this regularizer can be optimized, using recurrent back-propagation [18, 1, 25] to back-propagate through the few-shot optimization

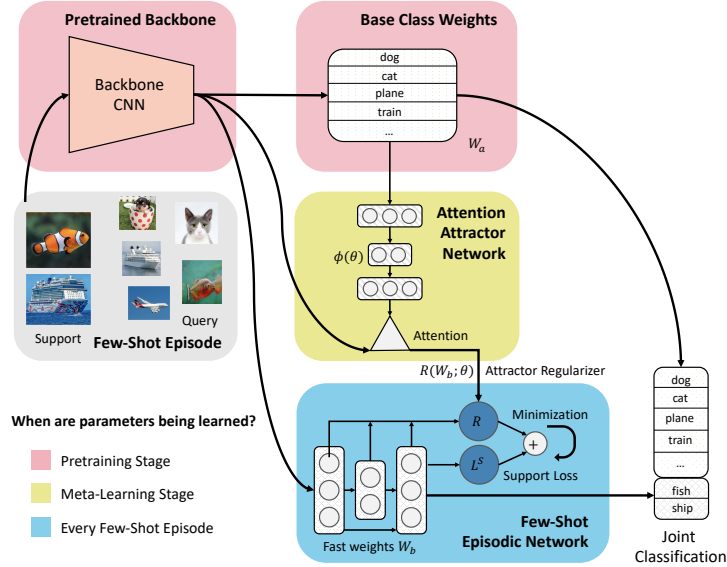

Figure 1: Our proposed attention attractor network for incremental few-shot learning. During pretraining we learn the base class weights $W_a$ and the feature extractor CNN backbone. In the meta-learning stage, a few-shot episode is presented. The support set only contains novel classes, whereas the query set contains both base and novel classes. We learn an episodic classifier network through an iterative solver, to minimize cross entropy plus an additional regularization term predicted by the attention attractor network by attending to the base classes. The attention attractor network is meta-learned to minimize the expected query loss. During testing an episodic classifier is learned in the same way.

stage. Finally, we show empirically that our proposed method can produce state-of-the-art results in incremental few-shot learning on *mini*-ImageNet [36] and *tiered*-ImageNet [29] tasks.

## 2   Related Work

Recently, there has been a surge in interest in few-shot learning [16, 36, 33, 17], where a model for novel classes is learned with only a few labeled examples. One family of approaches for few-shot learning, including Deep Siamese Networks [16], Matching Networks [36] and Prototypical Networks [33], follows the line of metric learning. In particular, these approaches use deep neural networks to learn a function that maps the input space to the embedding space where examples belonging to the same category are close and those belonging to different categories are far apart. Recently, [8] proposes a graph neural networks based method which captures the information propagation from the labeled support set to the query set. [29] extends Prototypical Networks to leverage unlabeled examples while doing few-shot learning. Despite their simplicity, these methods are very effective and often competitive with the state-of-the-art.

Another class of approaches aims to learn models which can adapt to the episodic tasks. In particular, [27] treats the long short-term memory (LSTM) as a meta learner such that it can learn to predict the parameter update of a base learner, e.g., a convolutional neural network (CNN). MAML [7] instead learns the hyperparameters or the initial parameters of the base learner by back-propagating through the gradient descent steps. [31] uses a read/write augmented memory, and [21] combines soft attention with temporal convolutions which enables retrieval of information from past episodes.

Methods described above belong to the general class of meta-learning models. First proposed in [32, 23, 35], meta-learning is a machine learning paradigm where the meta-learner tries to improve the base learner using the learning experiences from multiple tasks. Meta-learning methods typically learn the update policy yet lack an overall learning objective in the few-shot episodes. Furthermore, they could potentially suffer from short-horizon bias [41], if at test time the model is trained for

longer steps. To address this problem, [4] proposes to use fast convergent models like logistic regression (LR), which can be back-propagated via a closed form update rule. Compared to [4], our proposed method using recurrent back-propagation [18, 1, 25] is more general as it does not require a closed-form update, and the inner loop solver can employ any existing continuous optimizers.

Our work is also related to incremental learning, a setting where information is arriving continuously while prior knowledge needs to be transferred. A key challenge is *catastrophic forgetting* [20, 19], i.e., the model forgets the learned knowledge. Various memory-based models have since been proposed, which store training examples explicitly [28, 34, 5, 24], regularize the parameter updates [15], or learn a generative model [13]. However, in these studies, incremental learning typically starts from scratch, and usually performs worse than a regular model that is trained with all available classes together since it needs to learned a good representation while dealing with catastrophic forgetting.

Incremental few-shot learning is also known as low-shot learning. To leverage a good representation, [10, 37, 9] starts off with a pre-trained network on a set of base classes, and tries to augment the classifier with a batch of new classes that has not been seen during training. [10] proposes the squared gradient magnitude loss, which makes the learned classifier from the low-shot examples have a smaller gradient value when learning on all examples. [37] propose the prototypical matching networks, a combination of prototypical network and matching network. The paper also adds hallucination, which generates new examples. [9] proposes an attention based model which generates weights for novel categories. They also promote the use of cosine similarity between feature representations and weight vectors to classify images.

In contrast, during each few-shot episode, we directly learn a classifier network that is randomly initialized and solved till convergence, unlike [9] which directly output the prediction. Since the model cannot see base class data within the support set of each few-shot learning episode, it is challenging to learn a classifier that jointly classifies both base and novel categories. Towards this end, we propose to add a learned regularizer, which is predicted by a meta-network, the "attention attractor network". The network is learned by differentiating through few-shot learning optimization iterations. We found that using an iterative solver with the learned regularizer significantly improves the classifier model on the task of incremental few-shot learning.

# 3   Model

In this section, we first define the setup of incremental few-shot learning, and then we introduce our new model, the Attention Attractor Network, which attends to the set of base classes according to the few-shot training data by using the attractor regularizing term. Figure 1 illustrates the high-level model diagram of our method.

## 3.1   Incremental Few-Shot Learning

The outline of our meta-learning approach to incremental few-shot learning is: (1) We learn a fixed feature representation and a classifier on a set of base classes; (2) In each training and testing episode we train a novel-class classifier with our meta-learned regularizer; (3) We optimize our meta-learned regularizer on combined novel and base classes classification, adapting it to perform well in conjunction with the base classifier. Details of these stages follow.

**Pretraining Stage:**   We learn a base model for the regular supervised classification task on dataset $\{(x_{a,i}, y_{a,i})\}_{i=1}^{N_a}$ where $x_{a,i}$ is the $i$-th example from dataset $\mathcal{D}_a$ and its labeled class $y_{a,i} \in \{1, 2, ..., K\}$. The purpose of this stage is to learn both a good base classifier and a good representation. The parameters of the base classifier are learned in this stage and will be fixed after pretraining. We denote the parameters of the top fully connected layer of the base classifier $W_a \in \mathbb{R}^{D \times K}$ where $D$ is the dimension of our learned representation.

**Incremental Few-Shot Episodes:**   A few-shot dataset $\mathcal{D}_b$ is presented, from which we can sample few-shot learning episodes $\mathcal{E}$. Note that this can be the same data source as the pretraining dataset $\mathcal{D}_a$, but sampled episodically. For each $N$-shot $K'$-way episode, there are $K'$ novel classes disjoint from the base classes. Each novel class has $N$ and $M$ images from the support set $S_b$ and the query set $Q_b$ respectively. Therefore, we have $\mathcal{E} = (S_b, Q_b), S_b = (x_{b,i}^S, y_{b,i}^S)_{i=1}^{N \times K'}, Q_b = (x_{b,i}^Q, y_{b,i}^Q)_{i=1}^{M \times K'}$

where $y_{b,i} \in \{K+1, ..., K+K'\}$. $S_b$ and $Q_b$ can be regarded as this episodes training and validation sets. Each episode we learn a classifier on the support set $S_b$ whose learnable parameters $W_b$ are called the *fast weights* as they are only used during this episode. To evaluate the performance on a joint prediction of both base and novel classes, i.e., a $(K+K')$-way classification, a mini-batch $Q_a = \{(x_{a,i}, y_{a,i})\}_{i=1}^{M \times K}$ sampled from $\mathcal{D}_a$ is also added to $Q_b$ to form $Q_{a+b} = Q_a \cup Q_b$. This means that the learning algorithm, which only has access to samples from the novel classes $S_b$, is evaluated on the *joint* query set $Q_{a+b}$.

**Meta-Learning Stage:** In meta-training, we iteratively sample few-shot episodes $\mathcal{E}$ and try to learn the meta-parameters in order to minimize the joint prediction loss on $Q_{a+b}$. In particular, we design a regularizer $R(\cdot, \theta)$ such that the *fast weights* are learned via minimizing the loss $\ell(W_b, S_b) + R(W_b, \theta)$ where $\ell(W_b, S_b)$ is typically cross-entropy loss for few-shot classification. The meta-learner tries to learn meta-parameters $\theta$ such that the optimal *fast weights* $W_b^*$ w.r.t. the above loss function performs well on $Q_{a+b}$. In our model, meta-parameters $\theta$ are encapsulated in our attention attractor network, which produces regularizers for the fast weights in the few-shot learning objective.

**Joint Prediction on Base and Novel Classes:** We now introduce the details of our joint prediction framework performed in each few-shot episode. First, we construct an episodic classifier, *e.g.*, a logistic regression (LR) model or a multi-layer perceptron (MLP), which takes the learned image features as inputs and classifies them according to the few-shot classes.

During training on the support set $S_b$, we learn the *fast weights* $W_b$ via minimizing the following regularized cross-entropy objective, which we call the *episodic objective*:

$$L^S(W_b, \theta) = -\frac{1}{NK'} \sum_{i=1}^{NK'} \sum_{c=K+1}^{K+K'} y_{b,i,c}^S \log \hat{y}_{b,i,c}^S + R(W_b, \theta). \tag{1}$$

This is a general formulation and the specific functional form of the regularization term $R(W_b, \theta)$ will be specified later. The predicted output $\hat{y}_{b,i}^S$ is obtained via, $\hat{y}_{b,i}^S = \mathrm{softmax}([W_a^\top x_{b,i}, h(x_{b,i}; W_b^*)])$, where $h(x_{b,i})$ is our classification network and $W_b$ is the fast weights in the network. In the case of LR, $h$ is a linear model: $h(x_{b,i}; W_b) = W_b^\top x_{b,i}$. $h$ can also be an MLP for more expressive power.

During testing on the query set $Q_{a+b}$, in order to predict both base and novel classes, we directly augment the softmax with the fixed base class weights $W_a$, $\hat{y}_i^Q = \mathrm{softmax}([W_a^\top x_i, h(x_i; W_b^*)])$, where $W_b^*$ are the optimal parameters that minimize the regularized classification objective in Eq. (1).

### 3.2 Attention Attractor Networks

Directly learning the few-shot episode, e.g., by setting $R(W_b, \theta)$ to be zero or simple weight decay, can cause catastrophic forgetting on the base classes. This is because $W_b$ which is trained to maximize the correct novel class probability can dominate the base classes in the joint prediction. In this section, we introduce the Attention Attractor Network to address this problem. The key feature of our attractor network is the regularization term $R(W_b, \theta)$:

$$R(W_b, \theta) = \sum_{k'=1}^{K'} (W_{b,k'} - u_{k'})^\top \mathrm{diag}(\exp(\gamma))(W_{b,k'} - u_{k'}), \tag{2}$$

where $u_{k'}$ is the so-called *attractor* and $W_{b,k'}$ is the $k'$-th column of $W_b$. This sum of squared Mahalanobis distances from the attractors adds a bias to the learning signal arriving solely from novel classes. Note that for a classifier such as an MLP, one can extend this regularization term in a layer-wise manner. Specifically, one can have separate attractors per layer, and the number of attractors equals the number of output dimension of that layer.

To ensure that the model performs well on base classes, the attractors $u_{k'}$ must contain some information about examples from base classes. Since we can not directly access these base examples, we propose to use the *slow weights* to encode such information. Specifically, each base class has a learned attractor vector $U_k$ stored in the memory matrix $U = [U_1, ..., U_K]$. It is computed as, $U_k = f_\phi(W_{a,k})$, where $f$ is a MLP of which the learnable parameters are $\phi$. For each novel class $k'$ its classifier is regularized towards its attractor $u_{k'}$ which is a weighted sum of $U_k$ vectors. Intuitively

the weighting is an attention mechanism where each novel class attends to the base classes according to the level of interference, i.e. how prediction of new class $k'$ causes the forgetting of base class $k$.

For each class in the support set, we compute the cosine similarity between the average representation of the class and base weights $W_a$ then normalize using a softmax function

$$a_{k',k} = \frac{\exp\left(\tau A(\frac{1}{N}\sum_j h_j \mathbb{1}[y_{b,j}=k'], W_{a,k})\right)}{\sum_k \exp\left(\tau A(\frac{1}{N}\sum_j h_j \mathbb{1}[y_{b,j}=k'], W_{a,k})\right)}, \tag{3}$$

where $A$ is the cosine similarity function, $h_j$ are the representations of the inputs in the support set $S_b$ and $\tau$ is a learnable temperature scalar. $a_{k',k}$ encodes a normalized pairwise attention matrix between the novel classes and the base classes. The attention vector is then used to compute a linear weighted sum of entries in the memory matrix $U$, $u_{k'} = \sum_k a_{k',k} U_k + U_0$, where $U_0$ is an embedding vector and serves as a bias for the attractor.

Our design takes inspiration from attractor networks [22, 42], where for each base class one learns an "attractor" that stores the relevant memory regarding that class. We call our full model "dynamic attractors" as they may vary with each episode even after meta-learning. In contrast if we only have the bias term $U_0$, i.e. a single attractor which is shared by all novel classes, it will not change after meta-learning from one episode to the other. We call this model variant the "static attractor".

In summary, our meta parameters $\theta$ include $\phi$, $U_0$, $\gamma$ and $\tau$, which is on the same scale as as the number of paramters in $W_a$. It is important to note that $R(W_b, \theta)$ is convex w.r.t. $W_b$. Therefore, if we use the LR model as the classifier, the overall training objective on episodes in Eq. (1) is convex which implies that the optimum $W_b^*(\theta, S_b)$ is guaranteed to be unique and achievable. Here we emphasize that the optimal parameters $W_b^*$ are functions of parameters $\theta$ and few-shot samples $S_b$.

During meta-learning, $\theta$ are updated to minimize an expected loss of the query set $Q_{a+b}$ which contains both base and novel classes, averaging over all few-shot learning episodes,

---

**Algorithm 1** Meta Learning for Incremental Few-Shot Learning

**Require:** $\theta_0$, $\mathcal{D}_a$, $\mathcal{D}_b$, $h$
**Ensure:** $\theta$
1: $\theta \leftarrow \theta_0$;
2: **for** $t = 1 ... T$ **do**
3: $\quad \{(x_b^S, y_b^S)\}, \{(x_b^Q, y_b^Q)\} \leftarrow$ GetEpisode($\mathcal{D}_b$);
4: $\quad \{x_{a+b}^Q, y_{a+b}^Q\} \leftarrow$ GetMiniBatch($\mathcal{D}_a$) $\cup \{(x_b^Q, y_b^Q)\}$;
5:
6: $\quad$ **repeat**
7: $\quad\quad L^S \leftarrow \frac{1}{NK'}\sum_i y_{b,i}^S \log \hat{y}_{b,i}^S + R(W_b; \theta)$;
8: $\quad\quad W_b \leftarrow$ OptimizerStep($W_b, \nabla_{W_b} L^S$);
9: $\quad$ **until** $W_b$ converges
10: $\quad \hat{y}_{a+b,j}^Q \leftarrow$ softmax($[W_a^\top x_{a+b,j}^Q, h(x_{a+b,j}^Q; W_b)]$);
11: $\quad L^Q \leftarrow \frac{1}{2NK'}\sum_j y_{a+b,j}^Q \log \hat{y}_{a+b,j}^Q$;
12:
$\quad$ // Backprop through the above optimization via RBP
$\quad$ // A dummy gradient descent step
13: $\quad W_b' \leftarrow W_b - \alpha \nabla_{W_b} L^S$;
14: $\quad J \leftarrow \frac{\partial W_b'}{\partial W_b}; v \leftarrow \frac{\partial L^Q}{\partial W_b}; g \leftarrow v$;
15: $\quad$ **repeat**
16: $\quad\quad v \leftarrow J^\top v - \epsilon v; g \leftarrow g + v$;
17: $\quad$ **until** $g$ converges
18:
19: $\quad \theta \leftarrow$ OptimizerStep($\theta, g^\top \frac{\partial W_b'}{\partial \theta}$)
20: **end for**

---

$$\min_\theta \;\; \mathbb{E}_{\mathcal{E}}\left[L^Q(\theta, S_b)\right] = \mathbb{E}_{\mathcal{E}}\left[\sum_{j=1}^{M(K+K')}\sum_{c=1}^{K+K'} y_{j,c}\log \hat{y}_{j,c}(\theta, S_b)\right], \tag{4}$$

where the predicted class is $\hat{y}_j(\theta, S_b) = \text{softmax}\left([W_a^\top x_j, h(x_j; W_b^*(\theta, S_b))]\right)$.

### 3.3 Learning via Recurrent Back-Propagation

As there is no closed-form solution to the episodic objective (the optimization problem in Eq. 1), in each episode we need to minimize $L^S$ to obtain $W_b^*$ through an iterative optimizer. The question is how to efficiently compute $\frac{\partial W_b^*}{\partial \theta}$, *i.e.*, back-propagating through the optimization. One option is to unroll the iterative optimization process in the computation graph and use back-propagation through time (BPTT) [38]. However, the number of iterations for a gradient-based optimizer to converge can be on the order of thousands, and BPTT can be computationally prohibitive. Another way is to use

Table 1: Comparison of our proposed model with other methods

| Method | Few-shot learner | Episodic objective | Attention mechanism |
|--------|------------------|--------------------|--------------------|
| Imprint [26] | Prototypes | N/A | N/A |
| LwoF [9] | Prototypes + base classes | N/A | Attention on base classes |
| Ours | A fully trained classifier | Cross entropy on novel classes | Attention on learned attractors |

the truncated BPTT [39] (T-BPTT) which optimizes for $T$ steps of gradient-based optimization, and is commonly used in meta-learning problems. However, when $T$ is small the training objective could be significantly biased.

Alternatively, the recurrent back-propagation (RBP) algorithm [1, 25, 18] allows us to back-propagate through the fixed point efficiently without unrolling the computation graph and storing intermediate activations. Consider a vanilla gradient descent process on $W_b$ with step size $\alpha$. The difference between two steps $\Phi$ can be written as $\Phi(W_b^{(t)}) = W_b^{(t)} - F(W_b^{(t)})$, where $F(W_b^{(t)}) = W_b^{(t+1)} = W_b^{(t)} - \alpha \nabla L^S(W_b^{(t)})$. Since $\Phi(W_b^*(\theta))$ is identically zero as a function of $\theta$, using the implicit function theorem we have $\frac{\partial W_b^*}{\partial \theta} = (I - J_{F,W_b^*}^\top)^{-1} \frac{\partial F}{\partial \theta}$, where $J_{F,W_b^*}$ denotes the Jacobian matrix of the mapping $F$ evaluated at $W_b^*$. Algorithm 1 outlines the key steps for learning the episodic objective using RBP in the incremental few-shot learning setting. Note that the RBP algorithm implicitly inverts $(I - J^\top)$ by computing the matrix inverse vector product, and has the same time complexity compared to truncated BPTT given the same number of unrolled steps, but meanwhile RBP does not have to store intermediate activations.

**Damped Neumann RBP**   To compute the matrix-inverse vector product $(I - J^\top)^{-1}v$, [18] propose to use the Neumann series: $(I - J^\top)^{-1}v = \sum_{n=0}^{\infty} (J^\top)^n v \equiv \sum_{n=0}^{\infty} v^{(n)}$. Note that $J^\top v$ can be computed by standard back-propagation. However, directly applying the Neumann RBP algorithm sometimes leads to numerical instability. Therefore, we propose to add a damping term $0 < \epsilon < 1$ to $I - J^\top$. This results in the following update: $\tilde{v}^{(n)} = (J^\top - \epsilon I)^n v$. In practice, we found the damping term with $\epsilon = 0.1$ helps alleviate the issue significantly.

## 4   Experiments

We experiment on two few-shot classification datasets, *mini*-ImageNet and *tiered*-ImageNet. Both are subsets of ImageNet [30], with images sizes reduced to $84 \times 84$ pixels. We also modified the datasets to accommodate the incremental few-shot learning settings. [1]

### 4.1   Datasets

- ***mini*-ImageNet** Proposed by [36], *mini*-ImageNet contains 100 object classes and 60,000 images. We used the splits proposed by [27], where training, validation, and testing have 64, 16 and 20 classes respectively.

- ***tiered*-ImageNet** Proposed by [29], *tiered*-ImageNet is a larger subset of ILSVRC-12. It features a categorical split among training, validation, and testing subsets. The categorical split means that classes that belong to the same high-level category, e.g. "working dog" and "terrier" or some other dog breed, are not split between training, validation and test. This is a harder task, but one that more strictly evaluates generalization to new classes. It is also an order of magnitude larger than *mini*-ImageNet.

### 4.2   Experiment setup

We use a standard ResNet backbone [11] to learn the feature representation through supervised training. For *mini*-ImageNet experiments, we follow [21] and use a modified version of ResNet-10.

| Table 2: *mini*-ImageNet 64+5-way results | | | | | | Table 3: *tiered*-ImageNet 200+5-way results | | | | | |

| Model | 1-shot | | 5-shot | | | Model | 1-shot | | 5-shot | |
|---|---|---|---|---|---|---|---|---|---|---|
| | Acc. ↑ | Δ ↓ | Acc. ↑ | Δ ↓ | | | Acc. ↑ | Δ ↓ | Acc. ↑ | Δ ↓ |
| ProtoNet [33] | $42.73 \pm 0.15$ | -20.21 | $57.05 \pm 0.10$ | -31.72 | | ProtoNet [33] | $30.04 \pm 0.21$ | -29.54 | $41.38 \pm 0.28$ | -26.39 |
| Imprint [26] | $41.10 \pm 0.20$ | -22.49 | $44.68 \pm 0.23$ | -27.68 | | Imprint [26] | $39.13 \pm 0.15$ | -22.26 | $53.60 \pm 0.18$ | -16.35 |
| LwoF [9] | $52.37 \pm 0.20$ | -13.65 | $59.90 \pm 0.20$ | -14.18 | | LwoF [9] | $52.40 \pm 0.33$ | -8.27 | $62.63 \pm 0.31$ | -6.72 |
| Ours | $\mathbf{54.95} \pm 0.30$ | -11.84 | $\mathbf{63.04} \pm 0.30$ | **-10.66** | | Ours | $\mathbf{56.11} \pm 0.33$ | **-6.11** | $\mathbf{65.52} \pm 0.31$ | **-4.48** |

$\Delta$ = average decrease in acc. caused by *joint* prediction within base and novel classes ($\Delta = \frac{1}{2}(\Delta_a + \Delta_b)$) $\uparrow$ ($\downarrow$) represents higher (lower) is better.

For *tiered*-ImageNet, we use the standard ResNet-18 [11], but replace all batch normalization [12] layers with group normalization [40], as there is a large distributional shift from training to testing in *tiered*-ImageNet due to categorical splits. We used standard data augmentation, with random crops and horizonal flips. We use the same pretrained checkpoint as the starting point for meta-learning.

In the meta-learning stage as well as the final evaluation, we sample a few-shot episode from the $\mathcal{D}_b$, together with a regular mini-batch from the $\mathcal{D}_a$. The base class images are added to the query set of the few-shot episode. The base and novel classes are maintained in equal proportion in our experiments. For all the experiments, we consider 5-way classification with 1 or 5 support examples (i.e. shots). In the experiments, we use a query set of size $25 \times 2 = 50$.

We use L-BFGS [43] to solve the inner loop of our models to make sure $W_b$ converges. We use the ADAM [14] optimizer for meta-learning with a learning rate of 1e-3, which decays by a factor of 10 after 4,000 steps, for a total of 8,000 steps. We fix recurrent backpropagation to 20 iterations and $\epsilon = 0.1$.

We study two variants of the classifier network. The first is a logistic regression model with a single weight matrix $W_b$. The second is a 2-layer fully connected MLP model with 40 hidden units in the middle and tanh non-linearity. To make training more efficient, we also add a shortcut connection in our MLP, which directly links the input to the output. In the second stage of training, we keep all backbone weights frozen and only train the meta-parameters $\theta$.

## 4.3 Evaluation metrics

We consider the following evaluation metrics: 1) overall accuracy on individual query sets and the joint query set ("Base", "Novel", and "Both"); and 2) decrease in performance caused by *joint* prediction within the base and novel classes, considered separately ("$\Delta_a$" and "$\Delta_b$"). Finally we take the average $\Delta = \frac{1}{2}(\Delta_a + \Delta_b)$ as a key measure of the overall decrease in accuracy.

## 4.4 Comparisons

We implemented and compared to three methods. First, we adapted Prototypical Networks [33] to incremental few-shot settings. For each base class we store a base representation, which is the average representation (prototype) over all images belonging to the base class. During the few-shot learning stage, we again average the representation of the few-shot classes and add them to the bank of base representations. Finally, we retrieve the nearest neighbor by comparing the representation of a test image with entries in the representation store. In summary, both $W_a$ and $W_b$ are stored as the average representation of all images seen so far that belong to a certain class. We also compare to the following methods:

- **Weights Imprinting ("Imprint")** [26]: the base weights $W_a$ are learned regularly through supervised pre-training, and $W_b$ are computed using prototypical averaging.
- **Learning without Forgetting ("LwoF")** [9]: Similar to [26], $W_b$ are computed using prototypical averaging. In addition, $W_a$ is finetuned during episodic meta-learning. We implemented the most advanced variants proposed in the paper, which involves a class-wise attention mechanism. This model is the previous state-of-the-art method on incremental few-shot learning, and has better performance compared to other low-shot models [37, 10].

## 4.5 Results

We first evaluate our vanilla approach on the standard few-shot classification benchmark where no base classes are present in the query set. Our vanilla model consists of a pretrained CNN and a single-layer logistic regression with weight decay learned from scratch; this model performs on-par

Table 4: Ablation studies on *mini*-ImageNet

|  | 1-shot Acc. ↑ | Δ ↓ | 5-shot Acc. ↑ | Δ ↓ |
|---|---|---|---|---|
| LR | $52.74 \pm 0.24$ | -13.95 | $60.34 \pm 0.20$ | -13.60 |
| LR +S | $53.63 \pm 0.30$ | -12.53 | $62.50 \pm 0.30$ | -11.29 |
| LR +A | $\mathbf{55.31} \pm 0.32$ | **-11.72** | $63.00 \pm 0.29$ | -10.80 |
| MLP | $49.36 \pm 0.29$ | -16.78 | $60.85 \pm 0.29$ | -12.62 |
| MLP +S | $54.46 \pm 0.31$ | -11.74 | $62.79 \pm 0.31$ | -10.77 |
| MLP +A | $54.95 \pm 0.30$ | -11.84 | $\mathbf{63.04} \pm 0.30$ | **-10.66** |

Table 5: Ablation studies on *tiered*-ImageNet

|  | 1-shot Acc. ↑ | Δ ↓ | 5-shot Acc. ↑ | Δ ↓ |
|---|---|---|---|---|
| LR | $48.84 \pm 0.23$ | -10.44 | $62.08 \pm 0.20$ | -8.00 |
| LR +S | $55.36 \pm 0.32$ | -6.88 | $65.53 \pm 0.30$ | -4.68 |
| LR +A | $55.98 \pm 0.32$ | **-6.07** | $65.58 \pm 0.29$ | **-4.39** |
| MLP | $41.22 \pm 0.35$ | -10.61 | $62.70 \pm 0.31$ | -7.44 |
| MLP +S | $\mathbf{56.16} \pm 0.32$ | -6.28 | $\mathbf{65.80} \pm 0.31$ | -4.58 |
| MLP +A | $56.11 \pm 0.33$ | 6.11 | $65.52 \pm 0.31$ | -4.48 |

"+S" stands for static attractors, and "+A" for attention attractors.

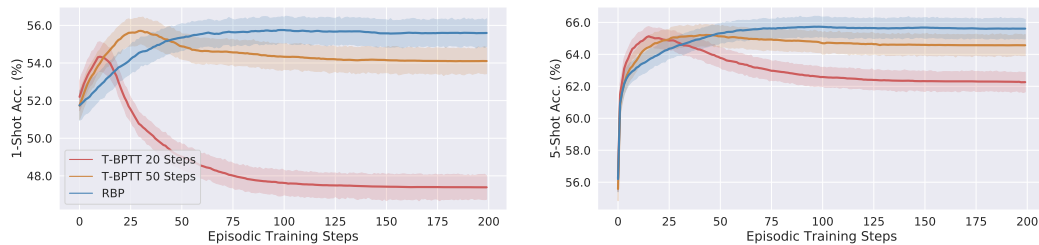

Figure 2: Learning the proposed model using truncated BPTT vs. RBP. Models are evaluated with 1-shot (left) and 5-shot (right) 64+5-way episodes, with various number of gradient descent steps.

with other competitive meta-learning approaches (1-shot $55.40 \pm 0.51$, 5-shot $70.17 \pm 0.46$). Note that our model uses the same backbone architecture as [21] and [9], and is directly comparable with their results. Similar findings of strong results using simple logistic regression on few-shot classification benchmarks are also recently reported in [6]. Our full model has similar performance as the vanilla model on pure few-shot benchmarks, and the full table is available in Supp. Materials.

Next, we compare our models to other methods on incremental few-shot learning benchmarks in Tables 2 and 3. On both benchmarks, our best performing model shows a significant margin over the prior works that predict the prototype representation without using an iterative optimization [33, 26, 9].

### 4.6 Ablation studies

To understand the effectiveness of each part of the proposed model, we consider the following variants:

- **Vanilla ("LR, MLP")** optimizes a logistic regression or an MLP network at each few-shot episode, with a weight decay regularizer.
- **Static attractor ("+S")** learns a fixed attractor center $u$ and attractor slope $\gamma$ for all classes.
- **Attention attractor ("+A")** learns the full attention attractor model. For MLP models, the weights below the final layer are controlled by attractors predicted by the average representation across all the episodes. $f_\phi$ is an MLP with one hidden layer of 50 units.

Tables 4 and 5 shows the ablation experiment results. In all cases, the learned regularization function shows better performance than a manually set weight decay constant on the classifier network, in terms of both jointly predicting base and novel classes, as well as less degradation from individual prediction. On *mini*-ImageNet, our attention attractors have a clear advantage over static attractors.

Formulating the classifier as an MLP network is slightly better than the linear models in our experiments. Although the final performance is similar, our RBP-based algorithm have the flexibility of adding the fast episodic model with more capacity. Unlike [4], we do not rely on an analytic form of the gradients of the optimization process.

**Comparison to truncated BPTT (T-BPTT)**  An alternative way to learn the regularizer is to unroll the inner optimization for a fixed number of steps in a differentiable computation graph, and then back-propagate through time. Truncated BPTT is a popular learning algorithm in many recent meta-learning approaches [2, 27, 7, 34, 3]. Shown in Figure 2, the performance of T-BPTT learned

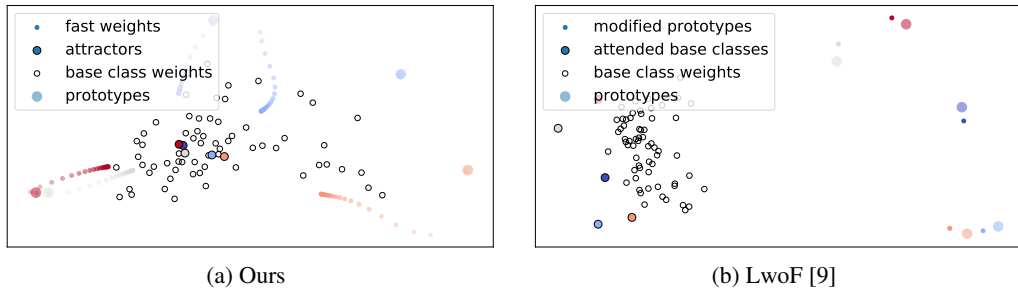

| (a) Ours | (b) LwoF [9] |

Figure 3: Visualization of a 5-shot 64+5-way episode using PCA. **Left:** Our attractor model learns to "pull" prototypes (large colored circles) towards base class weights (white circles). We visualize the trajectories during episodic training; **Right:** Dynamic few-shot learning without forgetting [9].

models are comparable to ours; however, when solved to convergence at test time, the performance of T-BPTT models drops significantly. This is expected as they are only guaranteed to work well for a certain number of steps, and failed to learn a good regularizer. While an early-stopped T-BPTT model can do equally well, in practice it is hard to tell when to stop; whereas for the RBP model, doing the full episodic training is very fast since the number of support examples is small.

**Visualization of attractor dynamics** We visualize attractor dynamics in Figure 3. Our learned attractors pulled the fast weights close towards the base class weights. In comparison, [9] only modifies the prototypes slightly.

**Varying the number of base classes** While the framework proposed in this paper cannot be directly applied on class-incremental continual learning, as there is no module for memory consolidation, we can simulate the continual learning process by varying the number of base classes, to see how the proposed models are affected by different stages of continual learning. Figure 4 shows that the learned regularizers consistently improve over baselines with weight decay only. The overall accuracy increases from 50 to 150 classes due to better representations on the backbone network, and drops at 200 classes due to a more challenging classification task.

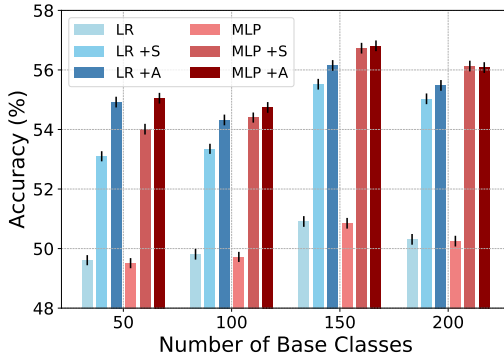

Figure 4: Results on *tiered*-ImageNet with {50, 100, 150, 200} base classes.

## 5 Conclusion

Incremental few-shot learning, the ability to jointly predict based on a set of pre-defined concepts as well as additional novel concepts, is an important step towards making machine learning models more flexible and usable in everyday life. In this work, we propose an attention attractor model, which regulates a per-episode training objective by attending to the set of base classes. We show that our iterative model that solves the few-shot objective till convergence is better than baselines that do one-step inference, and that recurrent back-propagation is an effective and modular tool for learning in a general meta-learning setting, whereas truncated back-propagation through time fails to learn functions that converge well. Future directions of this work include sequential iterative learning of few-shot novel concepts, and hierarchical memory organization.

**Acknowledgment** Supported by NSERC and the Intelligence Advanced Research Projects Activity (IARPA) via Department of Interior/Interior Business Center (DoI/IBC) contract number D16PC00003. The U.S. Government is authorized to reproduce and distribute reprints for Governmental purposes notwithstanding any copyright annotation thereon. Disclaimer: The views and conclusions contained herein are those of the authors and should not be interpreted as necessarily representing the official policies or endorsements, either expressed or implied, of IARPA, DoI/IBC, or the U.S. Government.

## Footnotes

[1]Code released at: `https://github.com/renmengye/inc-few-shot-attractor-public`

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
