[Supplementary Material]

# Supplementary Materials for "Incremental Few-Shot Learning with Attention Attractor Networks"

**Affiliation**
**Address**
`email`

## 1 Regular Few-Shot Classification

We include standard 5-way few-shot classification results in Table 1. As mentioned in the main text, a simple logistic regression model can achieve competitive performance on few-shot classification using pretrained features. Our full model shows similar performance on regular few-shot classification. This confirms that the learned regularizer is mainly solving the interference problem between the base and novel classes.

Table 1: Regular 5-way few-shot classification on *mini-ImageNet*. Note that this is purely few-shot, with no base classes. Applying logistic regression on pretrained features achieves performance on-par with other competitive meta-learning approaches. * denotes our own implementation.

| Model | Backbone | 1-shot | 5-shot |
|---|---|---|---|
| MatchingNets [8] | C64 | 43.60 | 55.30 |
| Meta-LSTM [5] | C32 | $43.40 \pm 0.77$ | $60.20 \pm 0.71$ |
| MAML [2] | C64 | $48.70 \pm 1.84$ | $63.10 \pm 0.92$ |
| RelationNet [7] | C64 | $50.44 \pm 0.82$ | $65.32 \pm 0.70$ |
| R2-D2 [1] | C256 | $51.20 \pm 0.60$ | $68.20 \pm 0.60$ |
| SNAIL [4] | ResNet | $55.71 \pm 0.99$ | $68.88 \pm 0.92$ |
| ProtoNet [6] | C64 | $49.42 \pm 0.78$ | $68.20 \pm 0.66$ |
| ProtoNet* [6] | ResNet | $50.09 \pm 0.41$ | $70.76 \pm 0.19$ |
| LwoF [3] | ResNet | $55.45 \pm 0.89$ | $\mathbf{70.92} \pm 0.35$ |
| LR | ResNet | $55.40 \pm 0.51$ | $70.17 \pm 0.46$ |
| Ours Full | ResNet | $\mathbf{55.75} \pm 0.51$ | $70.14 \pm 0.44$ |

## 2 Visualization of Few-Shot Episodes

We include more visualization of few-shot episodes in Figure 1, highlighting the differences between our method and "Dynamic Few-Shot Learning without Forgetting" [3].

## 3 Visualization of Attention Attractors

To further understand the attractor mechanism, we picked 5 semantic classes in *mini*-ImageNet and visualized their the attention attractors across 20 episodes, shown in Figure 2. The attractors roughly form semantic clusters, whereas the static attractor stays in the center of all attractors.

Figure 1: Visualization of 5-shot 64+5-way episodes on *mini*-ImageNet using PCA.

Figure 2: Visualization of example features and attractors using t-SNE. This plot shows a 5-way 5-shot episode on *mini*-ImageNet. 512-dimensional feature vectors and attractor vectors are projected to a 2-dim space. Color represents the label class of the example. The static attractor (teal) appears at the center of the attention attractors, which roughly form clusters based on the classes.

## 4 Dataset Statistics

In this section, we include more details on the datasets we used in our experiments.

### 4.1 Validation and testing splits for base classes

In standard few-shot learning, meta-training, validation, and test set have disjoint sets of object classes. However, in our incremental few-shot learning setting, to evaluate the model performance on the base class predictions, additional splits of validation and test splits of the meta-training set are required. Splits and dataset statistics are listed in Table 4. For *mini*-ImageNet, [3] released additional

Table 2: Full ablation results on 64+5-way *mini*-ImageNet

| | 1-shot | | | | 5-shot | | | |
|---|---|---|---|---|---|---|---|---|
| | Acc. ↑ | $\Delta\downarrow$ | $\Delta_a\downarrow$ | $\Delta_b\downarrow$ | Acc. ↑ | $\Delta\downarrow$ | $\Delta_a\downarrow$ | $\Delta_b\downarrow$ |
| LR | $52.74\pm0.24$ | -13.95 | -8.98 | -24.32 | $60.34\pm0.20$ | -13.60 | -10.81 | -15.97 |
| LR +S | $53.63\pm0.30$ | -12.53 | -9.44 | -15.62 | $62.50\pm0.30$ | -11.29 | -13.84 | -8.75 |
| LR +A | $\mathbf{55.31}\pm0.32$ | **-11.72** | -12.72 | -10.71 | $63.00\pm0.29$ | -10.80 | -13.59 | -8.01 |
| MLP | $49.36\pm0.29$ | -16.78 | -8.95 | -24.61 | $60.85\pm0.29$ | -12.62 | -11.35 | -13.89 |
| MLP +S | $54.46\pm0.31$ | -11.74 | -12.73 | -10.74 | $62.79\pm0.31$ | -10.77 | -12.61 | -8.80 |
| MLP +A | $54.95\pm0.30$ | -11.84 | -12.81 | -10.87 | $\mathbf{63.04}\pm0.30$ | **-10.66** | -12.55 | -8.77 |

Table 3: Full ablation results on 200+5-way *tiered*-ImageNet

| | 1-shot | | | | 5-shot | | | |
|---|---|---|---|---|---|---|---|---|
| | Acc. ↑ | $\Delta\downarrow$ | $\Delta_a\downarrow$ | $\Delta_b\downarrow$ | Acc. ↑ | $\Delta\downarrow$ | $\Delta_a\downarrow$ | $\Delta_b\downarrow$ |
| LR | $48.84\pm0.23$ | -10.44 | -11.65 | -9.24 | $62.08\pm0.20$ | -8.00 | -5.49 | -10.51 |
| LR +S | $55.36\pm0.32$ | -6.88 | -7.21 | -6.55 | $65.53\pm0.30$ | -4.68 | -4.72 | -4.63 |
| LR +A | $55.98\pm0.32$ | **-6.07** | -6.64 | -5.51 | $65.58\pm0.29$ | **-4.39** | -4.87 | -3.91 |
| MLP | $41.22\pm0.35$ | -10.61 | -11.25 | -9.98 | $62.70\pm0.31$ | -7.44 | -6.05 | -8.82 |
| MLP +S | $\mathbf{56.16}\pm0.32$ | -6.28 | -6.83 | -5.73 | $\mathbf{65.80}\pm0.31$ | -4.58 | -4.66 | -4.51 |
| MLP +A | $56.11\pm0.33$ | 6.11 | -6.79 | -5.43 | $65.52\pm0.31$ | -4.48 | -4.91 | -4.05 |

images for evaluating training set, namely "Train-Val" and "Train-Test". For *tiered*-ImageNet, we split out $\approx 20\%$ of the images for validation and testing of the base classes.

## 4.2 Novel classes

In *mini*-ImageNet experiments, the same training set is used for both $\mathcal{D}_a$ and $\mathcal{D}_b$. In order to pretend that the classes in the few-shot episode are novel, following [3], we masked the base classes in $W_a$, which contains 64 base classes. In other words, we essentially train for a 59+5 classification task. We found that under this setting, the progress of meta-learning in the second stage is not very significant, since all classes have already been seen before.

In *tiered*-ImageNet experiments, to emulate the process of learning novel classes during the second stage, we split the training classes into base classes ("Train-A") with 200 classes and novel classes ("Train-B") with 151 classes, just for meta-learning purpose. During the first stage the classifier is trained using Train-A-Train data. In each meta-learning episode we sample few-shot examples from the novel classes (Train-B) and a query base set from Train-A-Val.

## 200 Base Classes ("Train-A"):

n02128757, n02950826, n01694178, n01582220, n03075370, n01531178, n03947888,
n03884397, n02883205, n03788195, n04141975, n02992529, n03954731, n03661043,
n04606251, n03344393, n01847000, n03032252, n02128385, n04443257, n03394916,
n01592084, n02398521, n01748264, n04355338, n02481823, n03146219, n02963159,
n02123597, n01675722, n03637318, n04136333, n02002556, n02408429, n02415577,
n02787622, n04008634, n02091831, n02488702, n04515003, n04370456, n02093256,
n01693334, n02088466, n03495258, n02865351, n01688243, n02093428, n02410509,
n02487347, n03249569, n03866082, n04479046, n02093754, n01687978, n04350905,
n02488291, n02804610, n02094433, n03481172, n01689811, n04423845, n03476684,
n04536866, n01751748, n02028035, n03770439, n04417672, n02988304, n03673027,
n02492660, n03840681, n02011460, n03272010, n02089078, n03109150, n03424325,
n02002724, n03857828, n02007558, n02096051, n01601694, n04273569, n02018207,
n01756291, n04208210, n03447447, n02091467, n02089867, n02089973, n03777754,
n04392985, n02125311, n02676566, n02092002, n02051845, n04153751, n02097209,
n04376876, n02097298, n04371430, n03461385, n04540053, n04552348, n02097047,
n02494079, n03457902, n02403003, n03781244, n02895154, n02422699, n04254680,
n02672831, n02483362, n02690373, n02092339, n02879718, n02776631, n04141076,
n03710721, n03658185, n01728920, n02009229, n03929855, n03721384, n03773504,
n03649909, n04523525, n02088632, n04347754, n02058221, n02091635, n02094258,
n01695060, n02486410, n03017168, n02910353, n03594734, n02095570, n03706229,

Table 4: *mini*-ImageNet and *tiered*-ImageNet split statistics

| Classes | Purpose | *mini*-ImageNet | | | *tiered*-ImageNet | | |
|---------|---------|-----------------|--------|--------|-------------------|--------|---------|
| | | Split | N. Cls | N. Img | Split | N. Cls | N. Img |
| Base | Train | Train-Train | 64 | 38,400 | Train-A-Train | 200 | 203,751 |
| | Val | Train-Val | 64 | 18,748 | Train-A-Val | 200 | 25,460 |
| | Test | Train-Test | 64 | 19,200 | Train-A-Test | 200 | 25,488 |
| Novel | Train | Train-Train | 64 | 38,400 | Train-B | 151 | 193,996 |
| | Val | Val | 16 | 9,600 | Val | 97 | 124,261 |
| | Test | Test | 20 | 12,000 | Test | 160 | 206,209 |

n02791270, n02127052, n02009912, n03467068, n02094114, n03782006, n01558993,
n03841143, n02825657, n03110669, n03877845, n02128925, n02091032, n03595614,
n01735189, n04081281, n04328186, n03494278, n02841315, n03854065, n03498962,
n04141327, n02951585, n02397096, n02123045, n02095889, n01532829, n02981792,
n02097130, n04317175, n04311174, n03372029, n04229816, n02802426, n03980874,
n02486261, n02006656, n02025239, n03967562, n03089624, n02129165, n01753488,
n02124075, n02500267, n03544143, n02687172, n02391049, n02412080, n04118776,
n03838899, n01580077, n04589890, n03188531, n03874599, n02843684, n02489166,
n01855672, n04483307, n02096177, n02088364.

**151 Novel Classes ("Train-B"):**

n03720891, n02090379, n03134739, n03584254, n02859443, n03617480, n01677366,
n02490219, n02749479, n04044716, n03942813, n02692877, n01534433, n02708093,
n03804744, n04162706, n04590129, n04356056, n01729322, n02091134, n03788365,
n01739381, n02727426, n02396427, n03527444, n01682714, n03630383, n04591157,
n02871525, n02096585, n02093991, n02013706, n04200800, n04090263, n02493793,
n03529860, n02088238, n02992211, n03657121, n02492035, n03662601, n04127249,
n03197337, n02056570, n04005630, n01537544, n02422106, n02130308, n03187595,
n03028079, n02098413, n02098105, n02480855, n02437616, n02123159, n03803284,
n02090622, n02012849, n01744401, n06785654, n04192698, n02027492, n02129604,
n02090721, n02395406, n02794156, n01860187, n01740131, n02097658, n03220513,
n04462240, n01737021, n04346328, n04487394, n03627232, n04023962, n03598930,
n03000247, n04009552, n02123394, n01729977, n02037110, n01734418, n02417914,
n02979186, n01530575, n03534580, n03447721, n04118538, n02951358, n01749939,
n02033041, n04548280, n01755581, n03208938, n04154565, n02927161, n02484975,
n03445777, n02840245, n02837789, n02437312, n04266014, n03347037, n04612504,
n02497673, n03085013, n02098286, n03692522, n04147183, n01728572, n02483708,
n04435653, n02480495, n01742172, n03452741, n03956157, n02667093, n04409515,
n02096437, n01685808, n02799071, n02095314, n04325704, n02793495, n03891332,
n02782093, n02018795, n03041632, n02097474, n03404251, n01560419, n02093647,
n03196217, n03325584, n02493509, n04507155, n03970156, n02088094, n01692333,
n01855032, n02017213, n02423022, n03095699, n04086273, n02096294, n03902125,
n02892767, n02091244, n02093859, n02389026.