[Reviews · NeurIPS 2019]

Reviewer 1



The paper is nicely written, motivates incremental few-shot learning problem, and makes a couple of interesting architectural/algorithmic contributions. The results are interesting/convincing, and the ablation study helps to better understand the properties of the prosed method. I believe the paper would be of interest to the community but more clarifications are necessary. === Comments and questions: - Inconsistencies in softmaxes. From 3.1, it looks like the softmaxes computed used for training fast and slow weights have different normalization constants (see line 135 and line 139), and hence the logits for b-classes might have entirely different scales than logits for a-classes. I feel that the reason why you need the proposed attractor-based regularization in the first place is to compensate for this scaling issue. Alternatively, you can use the same softmax from line 139 when computing the loss in eq (1) to avoid this discrepancy (since the base classes are available, W_a is pretrained and fixed, this should be possible). Why not do that? I would like to see a comparison with the vanilla architecture (i.e., no attractor-based regularization) that uses a consistent softmax normalization. - Computational considerations. RBP requires the inverse of the Jacobian wich scales cubically. What is the computational overhead for this method? How would it scale with the increased number of classes? Analysis and discussion of this are necessary. - Results. The authors mention that tiered-ImageNet is a harder task (which intuitively makes sense), but somehow results on that dataset are better than on the mini-ImageNet. How would you explain that? - Will the code for reproducing results be released? === Minor: - line 124: I believe \theta_E denotes parameters of the feature extractor, but it has not been properly defined.

Reviewer 2



Overall a very nice and interesting read. In terms of originality I believe the proposed method to be sufficiently novel and at no point felt this was merely an incremental improvement. In terms of significance it should be said that I feel this idea to be fairly specific to the incremental classification setting and wouldn't be general enough to be directly applicable in another domain (e.g. RL). However, I still believe this work should be accepted and would expect recognition within the domain. With regards to the clarity of the submission, I believe sections 3.1 and 3.2 could be improved. Detailed comments below: Introduction: L36: "We optimize a regularizer that reduces catastrophic forgetting" - Perhaps it would be a good idea to delineate this from many of the other works on regularization-based methods to reduce catastrophic forgetting where the regularizer isn't learnt? Examples are [1] or [2]. L37 "can be thought of as a memory of the base classes, adapted to the new classes". This is very unclear and not particularly helpful in the introduction. I could only make sense of this sentence after reading Section 3. Figure 1: Very helpful, thank you for providing this. Section 2: Nice overview of related work. Perhaps some more discussion on work to tackle the catastrophic forgetting problem would be useful here. Section 3: L112: "there are K' novel classes disjoint from the base classes" - When we use the same datset D_a also used during pretraining, this seems only possible when data-augmentation is introduced (as the authors explain in Section 4 (L239)). It would be good to already mention this here (possibly with a footnote). Also, I understand data-augmentation as training on the union of original data and a distroted version thereof. In order to ensure that the K' classes are indeed disjoint, are the authors ensuring that during episodic sampling from D_a there is always some form of distortion applied? If for instance, during sampling of random rotations we choose between {90, 180, 270, 360), one could run into the risk of training on identical data already used during pre-training. L120: "from from" -> "from" L126: Where do parameters \theta_E fit in Figure 1? They appear as arguments to R_(W_b, \theta_E) but never appear in the definition of equation (2). This is not made clear until line 172 which was rather confusing. Also the subscript E seems like a stranger letter to choose. L139: W_a are called "slow weights" (also in L154) whereas they have previously been referred to as "Base class Weights" (Figure 1). I found myself repeatedly looking at Figure 1 to keep track of what was happening in the model. Using consistent notation would have made this a lot easier. Section 3.3: Very clear. Section 4: L245: Missing whitespace after the full-stop. L248: "RBP" -> Recurent back-propagation as readers might be unfamiliar with the abbrevation and might want to only briefly skim the experimental section. Section 4.3: 2) I don't think I understand what \delta_a and \delta_b are supposed to be. Section 4.6: Nice to see the ablation study I was hoping for when reading Section 3. Interesting also to see that in some cases a simple LR model for W_b works better or just as well. Also great to see the comparison between RBP and T-BPTT Figure 3 is really nice. [1] Kirkpatrick, James, et al. "Overcoming catastrophic forgetting in neural networks." Proceedings of the national academy of sciences 114.13 (2017): 3521-3526. [2] Nguyen, Cuong V., et al. "Variational continual learning." arXiv preprint arXiv:1710.10628 (2017).

Reviewer 3



Originality: The proposed Attention Attractor Network is novel. Quality: The intuition of using the attention attractor network is not that sound. The motivation of the model is to prevent forgetting base classes. But the method (line 160-166) takes the cosine similarity between the average representation of novel classes and the base class weights. This is purely intuitive. Clarity: There some places not very clear. Significance: the empirical results seem significant.

[Author Response · NeurIPS 2019]

**Author response for "Incremental Few-Shot Learning with Attention Attractor Networks"**

We thank all reviewers for their time and insightful comments. We address individual comments below.

**To R1 on the normalization constant:** Thank you for pointing this out. In our experiments we actually used the same normalization constant (same softmaxes) for both support set and query set, both having $W_a$. Predictions for the base classes are simply ignored when calculating the loss for the support set. We will fix the paper to clarify this.

**To R1 on *mini-* vs. *tiered*-ImageNet:** *tiered*-ImageNet contains many more images, so this allows the network to be pretrained with a better feature representation, and we are also able to fit it with a larger and deeper network (ResNet-18 vs. ResNet-10). The challenge in tiered ImageNet lies in the split on higher level class categories, so there is a domain shift in training, validation and testing.

**To R1 on computation overhead:** Thank you for asking. We are not directly inverting the matrix but computing the inverse matrix vector product. Second, we are using truncated RBP, which computes a low-rank approximation of the matrix inverse, i.e. doing Jacobian transpose vector product for a fixed number of steps (see Line 15 in Alg. 1). In the experiments, we used 20 steps, so this has the same time complexity as truncated BPTT for 20 steps, while RBP saves memory by not having to store intermediate activations. We will add these clarifications in the paper.

**To R1 on code release:** We will release the code that can reproduce the experiments if the paper gets accepted. The release is expected to happen before the conference.

**To R2 on catastrophic forgetting literature:** Thank you for providing the references. We will cite them and include them in our discussion of related work. [1] uses the diagonal approximation of the Fisher matrix to prevent the parameter from drifting too far. Since our backbone network is frozen, this wouldn't be an issue in our setting, however, it could be complementary to our method if we choose to finetune the backbone network as well. [2] stores a few data points of old tasks in the "Coreset" so that they can estimate the variational distribution, whereas in our case, the regularizer is learned through meta-learning.

**To R2 on disjoint classes & data augmentation:** Data augmentation is only applied during pretraining, and for simplicity not applied during meta-learning. To ensure that in meta-learning novel classes do not overlap with base classes, we mask out the base classes that are used in the few-shot episode.

**To R2 on $\theta_E$ in Figure 1:** Thanks for the suggestion. We will modify Figure 1 to match with the notations. The meta parameters $\theta_E$ is enclosed in the teal part of the figure, which is learned during meta-learning stage, i.e. for mini ImageNet, there are 64 training classes, and the query set will be 59+5 classes, where there are 5 "fake" novel classes.

**To R2 on other notation issues:** Thanks for pointing them out. We will adopt your suggestions and change the notation to ensure consistency.

**To R2 on $\Delta_a$ and $\Delta_b$:** These two measure how much drop in accuracy is caused by jointly classifying base and novel classes. We first record the accuracy by classifying base classes alone (i.e. standard classification problem), and novel classes alone (i.e. standard few-shot problem), and then observe the accuracy of base and novel classes in a mixed query set. Lastly, we take the difference of these two accuracies to be the drop in performance ($\Delta_a$ and $\Delta_b$).

**To R3 on motivation of attention mechanism:** The motivation is to assess the similarity between the novel classes and the base classes, so that during the learning of novel classes, the system learns to differentiate from the base classes and prevent interference. The attention vector is used to retrieve an attractor to regularize the episodic training.

**To R3 on clarity:** Could you comment on the specific places that you feel are not clearly written? We will revise these sections in the next version.

**To R3 on OptimizerStep:** Line 12-17 is the part of the algorithm that computes the gradients of the meta-parameters $\theta_E$. "OptimizerStep" is an external function that performs optimization given some gradient direction (e.g. gradient descent, momentum, Adam, etc.).

**To R3 on T-BPTT:** The reason why T-BPTT fails is that it only optimizes for a short horizon (e.g. 20-100 steps of gradient descent). When we train the episodic objective till convergence, T-BPTT is no longer trained for that, so the accuracy goes down for longer iterations.

[Meta-Review · NeurIPS 2019]

The authors proposed a new attention attractor for incremental few-shot learning where base classifier is trained offline with enough number of data and additional extra novel classes are added later, each with only a few labeled examples. The setting is important and interesting. The idea is novel and results are overall quite strong. There are some concerns regarding the clarity; this should be revised in the final version.